# High Tg/HDL-Cholesterol Ratio Highlights a Higher Risk of Metabolic Syndrome in Children and Adolescents with Severe Obesity

**DOI:** 10.3390/jcm11154488

**Published:** 2022-08-01

**Authors:** Giorgio Radetti, Graziano Grugni, Fiorenzo Lupi, Antonio Fanolla, Diana Caroli, Adele Bondesan, Alessandro Sartorio

**Affiliations:** 1Marienklinik, 39100 Bolzano, Italy; 2Experimental Laboratory for Auxo-Endocrinological Research, Division of Auxology, Istituto Auxologico Italiano, IRCCS, 28824 Piancavallo-Verbania, Italy; g.grugni@auxologico.it (G.G.); d.caroli@auxologico.it (D.C.); a.bondesan@auxologico.it (A.B.); sartorio@auxologico.it (A.S.); 3Newborn Intensive Care Unit, Regional Hospital of Bolzano, 39100 Bolzano, Italy; fiorenzolupi@gmail.com; 4Observatory for Health Provincial Government South Tyrol, 39100 Bolzano, Italy; antonio.fanolla@provincia.bz.it

**Keywords:** metabolic syndrome, children, adolescents, indexes

## Abstract

Few data are currently available on the reliability of the different anthropometric, instrumental and biochemical indexes in recognizing the presence of metabolic syndrome (MetS) in children and adolescents with severe obesity. Therefore, the objective of our study was to find out the simplest and most accurate predictive index of MetS in this population at-risk. In 1065 children and adolescents (563 f, 502 m), aged 14.6 ± 2.1 years (range 10–17), with severe obesity [BMI-SDS 3.50 ± 0.36 (range 3.00–5.17)], the following indexes were evaluated: BMI, BMI-SDS, Tri-Ponderal Mass Index, Waist-to-Height ratio, TG/HDL-Cholesterol ratio, Cardiometabolic Index (CMI), and Visceral Adiposity Index (VAI). For each subject, all the components of MetS, defined according to the IDF criteria, were determined. Overall, the presence of MetS was found in 324 patients (30.4%), 167 males (33.3%) and 157 females (27.9%). According to the ROC analysis, three indexes (VAI, CMI and TG/HDL-Cholesterol ratio), performed significantly better than the other ones in identifying MetS, with no difference among them. In conclusion, the TG/HDL ratio, which just needs the evaluation of two simple biochemical parameters, offers the same accuracy as other more sophisticated indexes in recognizing MetS in children and adolescents with severe obesity, thus making it the best predictor to be easily used.

## 1. Introduction

Pediatric obesity is one of the major public health concerns of the current century, with an increasing prevalence of severe forms over the last decades [1,2,3]. This problem has further worsened due as a result of the recent COVID 19 pandemic, which has led to a significant increase of obesity as a result of its negative influence on lifestyles [4]. Since severe obesity is closely associated with metabolic syndrome (MetS) [5], which is a strong risk factor for cardiovascular diseases and type 2 diabetes mellitus (T2DM), this situation has consequently had a significant impact on health services as well as on socioeconomic system [6,7]. For this reason, it is necessary to have tools available to identify those patients with obesity at risk of developing MetS at an early stage. For this purpose, several indexes have been proposed over the years with the aim of the early detection of the presence of MetS in obese subjects. Body Mass Index (BMI) is the most commonly used surrogate index in clinical practice, even if it does not discriminate fat mass from lean mass or consider fat distribution. Given that the amount of fat mass and its distribution are believed to be the most important metabolic risk factors [8,9], other indexes based on these parameters have been subsequently developed. In this respect, several authors have proposed indexes which define adiposity (Fat Mass Index (FMI) [10], Tri-Ponderal Mass Index (TMI) [11], or that assess body fat distribution [Waist Circumference (WC) [12], Visceral Adiposity Index (VAI) [13], Waist-to-Height ratio (WtHR) [14], or Body Mass Fat Index (BMFI) [15]. In addition, the Cardiometabolic Index (CMI), consisting of indexes of adiposity and blood lipids included in the criteria of MetS, and the triglyceride to high density-lipoprotein cholesterol (TG/HDL-C) ratio were both reported as accurate measures for predicting MetS [16,17,18]. So far, no consensus has been reached on which index is the best one. The objective is to find the simplest and least expensive tool that can identify the probability of having MetS in obese subjects, avoiding unnecessary investigation. In this context, we have previously compared indexes requiring or not instrumental support in a large group of children and adolescents, with different degrees of obesity [19]. We demonstrated that the indexes that considered body composition, as assessed by Bioelectrical Impedance Analysis (BIA), did not perform better than BMI for detecting MetS. More recently, we have demonstrated that the evaluation of body composition throughout BIA did not add any additional value to simple anthropometric measures, such as BMI or WtHR, in identifying the risk of MetS in a large number of obese women [19].

Little data is currently available on the reliability of the different indexes in recognizing the presence of MetS in severe childhood obesity. Vizzuso et al. [20], evaluating a cohort of 637 obese children and adolescent, including 215 patients with BMI z-score > 3, found a higher VAI in patients with MetS compared to the subgroup without MetS. Interestingly, these authors demonstrated that VAI was the best predictor of MetS when compared to BMI z-score, A Body Shape Index (ABSI) z-score and WtHR z-score.

With this background, in the present study we compared the accuracy of a group of indexes, which only consider anthropometric measures and simple biochemical data, in a large group of severely obese children and adolescents. The objective was to find out a simple and accurate index to be used for the prediction of MetS in this population at-risk, in which its early detection is mandatory in order to avoid serious metabolic and cardiovascular complications.

## 2. Materials and Methods

### 2.1. Study Population

A retrospective cohort study based on 1065 children and adolescents with severe obesity [563 females, 502 males, aged 14.6 ± 2.1 (range 10–17) years, height standard deviation score (SDS) 0.35 ± 0.96 (range −1.58/3.97) and BMI SDS 3.50 ± 0.36 (range 3.00–5.17)], consecutively hospitalized at the Division of Auxology, Istituto Auxologico Italiano, Piancavallo, Verbania, Italy between January 2018 and January 2021 was performed. All but 51 (4.8%) patients were Caucasian. All of the subjects suffered from essential obesity, other genetic, organic, endocrine or iatrogenic forms having been excluded. None of them had received treatment with any drug known to interfere with metabolism, such as oral contraceptives and anorectic, antihypertensive, lipid lowering, or insulin sensitizer drugs during the previous 12 months. Pubertal development was assessed according to Tanner classification [21]. One hundred and fifteen patients were prepubertal (stage 1), 488 were pubertal (stage 2–4), and 462 were fully developed (stage 5). The study protocol was approved by the Ethical Committee of the Istituto Auxologico Italiano (ref. no. 01C822; acronym: METOBIP). At the admission to our Institute, written informed consent was obtained from the parents or legal guardians for the use of all biochemical and anthropometric parameters collected during hospitalization, as well as written assent from children and adolescents. The study was performed in accordance with the Declaration of Helsinki and with the 2005 Additional Protocol to the European Convention of Human Rights and Medicine concerning Biomedical Research.

### 2.2. Anthropometric Data

All subjects underwent body measurements wearing light underwear, in fasting conditions after voiding. Physical examination included the determination of height, weight, and waist circumference (WC) by the same trained operators, according to the Anthropometric Standardization Reference Manual [22]. Standing height was determined by a Harpenden Stadiometer (Holtain Limited, Crymych, Dyfed, UK). Body weight was measured to the nearest 0.1 kg using an electronic scale (Ro WU 150, Wunder Sa.bi., Trezzo sull’Adda, Italy). WC was determined in standing position midway between the lowest rib and the top of the iliac crest after gentle expiration, with a non-elastic flexible tape measure.

### 2.3. Blood Pressure Measurements

Diastolic and systolic blood pressure (BP) were measured to the nearest 2 mmHg in the supine position after 5 min rest, using a standard mercury sphygmomanometer with an appropriately sized cuff. The average of three measurements on different days was used.

### 2.4. Laboratory Analyses

Baseline blood samples were drawn by venipuncture after a 12-h overnight fast. Measurements of glycemia, high-density lipoprotein cholesterol (HDL-C), and triglycerides (TG) were performed by standard enzymatic methods (Roche Diagnostics, Mannheim, Germany).

### 2.5. Definitions

Severe obesity was considered to be a BMI ≥ 3.00 SDS (>99th percentile) [23,24], using the World Health Organization (WHO) reference charts [25]. According to the International Diabetes Federation (IDF) criteria [26], MetS was defined in the presence of abdominal obesity [WC ≥ 90th percentile for ages < 16 years [27], and 94 cm for males and 80 cm for female for ages > 16 years] plus two or more of the following four parameters: high systolic BP and/or diastolic BP, high triglycerides, low HDL-C, and elevated fasting glucose. Hypertension was defined as values of systolic BP ≥ 130 mmHg or diastolic BP ≥ 80 mmHg. Hypertriglyceridemia was defined in the presence of triglycerides values ≥ 150 mg/dL (≥1.7 mmol/L). A reduced HDL-C level was defined with values < 40 mg/dL (1.03 mmol/L) for males and females for ages < 16 years, and <40 mg/dL for males and <50 mg/dL (1.29 mmol/L) for females for ages > 16 years. Elevated fasting glucose was defined as glycemia ≥ 5.6 mmol/L (or ≥100 mg/dL).

The following indexes were calculated according to the following formulas:

BMI [28]: weight (kg)/height in m^2^

BMI SDS [28]: (BMI-mean BMI (for age and sex))/SD

TMI [11]: mass (kg)/height (m^3^)

WtHR [14]: WC (cm)/height (cm)

TG/HDL-C ratio [29]: TG (mmol/L)/HDL-C (mmol/L)

CMI [30]: WtHR × TG/HDL-C ratio

VAI [13]:

males: (WC/39.68 + (1.88 × BMI)) × (TG/1.03) × (1.31/HDL)

females: (WC/36.58 + (1.89 × BMI)) × (TG/0.81) × (1.52/HDL)

### 2.6. Statistical Analysis

The data (VAI, TG/HDL, CMI) were first scrutinized for outliers using a cutoff of 4.5 SDS. No patient was excluded on this basis. A Shapiro–Wilk test was used to assess normality of each continuous variable; all tested variables were non-normally distributed. The analysis of the distribution graphs confirmed that it was not possible to make assumptions of normality. To explore the data, preliminary analyses by gender and by having MetS or not having it were performed. Continuous data were presented as median (interquartile range). Median values were tested for statistical significance using a two-tailed Wilcoxon test and a nonparametric one-way ANOVA. Spearman correlation coefficients were calculated to assess the relationship between TG/HDL-C ratio and metabolic characteristics.

An adjusted receiver operating characteristic (ROC) analysis using clinical cut points for metabolic risks was performed to determinate the odds ratio of cardiometabolic risk factors across the TG/HDL-C ratio. ROC curves were generated in order to obtain the values of area under the curve (AUC), with sensitivity, specificity, and 95% CI, for the age-adjusted standardized body composition indexes as predictors of MetS [31]. To identify the optimal cut-off, the Youden index [32] was calculated. In addition, positive likelihood ratio (PLR), negative likelihood ratio (NLR), positive predictive value (PPV), and negative predictive value (NPV) were examined.

In order to calculate the growth pattern of the body-composition index a quantile regression was used [33] as an alternative for the LMS Method [34], with body composition indexes as a response, fitted with a parametric model which involved age. The standardized residuals were retained to represent age-adjusted values.

The significance threshold was set at *p* < 0.05. The data were analyzed using SAS Enterprise Guide 4.3 (SAS Institute Inc., Cary, NC, USA).

## 3. Results

The clinical and biochemical data, together with the different indexes in boys and girls, are shown in Table 1. According to the IDF criteria for MetS, all but two females fulfilled the criteria for abdominal obesity. High triglycerides values were found in 119 patients (59 males) (11.2%), while 483 had reduced HDL-C levels (222 males) (45.3%). High BP was detected in 535 subjects (291 males) (50.2%). Raised fasting glucose was found in eight individuals (4 males) (0.7%). Several significant differences were found between the two genders, while insulin, HOMA-IR, LDL-C and WtHR were comparable. Overall, boys showed a worse metabolic profile, higher WC and blood pressure than girls, although the latter had a higher BMI. Boys had lower VAI and TMI and higher TG/HDL-C ratio and CMI than girls.

In Table 2, the same parameters are reported according to the presence or absence of MetS. Overall, the presence of MetS was found in 324 patients (30.4%), 167 males (33.3%) and 157 females (27.9%) (not significant). As expected, the metabolic profile was significantly worse in the MetS group, with the only exception being glucose level, which was not different between the two subgroups.

In Figure 1 and Figure 2 the ROC curves in boys and girls are reported, respectively, showing the best performance of VAI, CMI and TG/HDL-C ratio in identifying MetS. No difference was found among these indexes (Appendix A Appendix A).

In Table 3, the sensitivity, specificity, positive likelihood ratio (PLR), negative likelihood ratio (NLR), positive predictive value (PPV), and negative predictive value (NPV) of VAI, CMI and TG/HDL-C ratio are reported. In boys, no difference among the three indexes for all parameters was found. Otherwise, in girls, CMI showed, on one hand, less sensitivity compared to the other indexes, and on the other hand the best specificity positive predictive values and the best positive likelihood ratio. The PLR of the three indexes were useful as predictive values for MetS in boys, while in girls only CMI was found to be useful. The negative likelihood ratio was similar in all subjects.

Consequently, the TG/HDL-C ratio was chosen as the preferred index due to its simplicity, because it did not require any additional anthropometric measures as compared to VAI and CMI.

In Table 4, the correlation between TG/HDL-C ratio and the metabolic factors in boys and girls is reported. A strong correlation with the clinical and biochemical parameters of MetS was detected in both sexes, except for serum glucose level.

In Table 5 the odds ratio between TG/HDL-C ratio with the different cardiometabolic risk factors are reported. In both sexes, a strong association between TG/HDL-C ratio and high triglycerides and low HDL-C was found.

In Figure 3 the trend of TG/HDL-C ratio, according to age and sex, is shown. The mean difference of the cut-off values calculated at all ages between the two sexes was always significant (*p* < 0.05).

## 4. Discussion

A substantial increase in the rates of severe obesity in children and adolescents has been demonstrated over the past decades, and this phenomenon appears to be nearly universal and not limited to a particular region [35]. The prevalence of severe obesity among subjects aged 2–19 years in United States, from the years 1971–1974 to the years 2017–2018, raised from 1.0% to 6.1% [36]. More recently, however, the assessment of changes over time for nine European countries showed a decrease of severe childhood obesity (starting from 2007–2008) in three of them, including Italy [2]. Despite this reduction, the overall levels of obesity in Italy remain among the highest in Europe [37], and in 2010 it was estimated that around 50,000 children eight to nine years of age suffered from severe obesity according to WHO criteria [38].

The significance of severe obesity on associated morbidity at an early age should be readily investigated. In our cohort of 1065 subjects we found the presence of MetS in 324 of them, with a prevalence almost double in comparison to previously reported data (30.4% vs. 16.7%) using the same criteria to define severe obesity [20]. This discrepancy might be related to the different clinical characteristics of the study groups, including age-range and number of subjects. In this context, a simple, reliable, and minimally invasive index should be necessary in order to screen for the presence of MetS in a population of severe obese children and adolescents at high risk for metabolic and cardiac disease. For this purpose, in this study we have selected indexes based on anthropometric data only or with the simple addition of routine biochemical parameters, such as triglycerides and HDL cholesterol, avoiding those indexes which require an instrumental support to evaluate body composition. We found that three indexes (i.e., VAI, CMI and TG/HDL-C ratio), performed significantly better than the other ones (i.e., BMI, BMI SDS, WtHR and TMI), with no difference among them. As a result, we only considered the best performing indexes. In this regard, we were quite surprised by the absence of differences between the selected ones, since VAI and CMI take into consideration not only triglycerides and HDL cholesterol levels but also measures of body excess and fat distribution, the latter being actually considered a marker for a higher risk of cardiometabolic diseases. A likely explanation might be that we studied a very peculiar population, i.e., a group of extremely obese adolescents with a BMI SDS of more than 3 SDS. Presumably because of this condition, these subjects already have a consolidated deranged metabolic milieu and this situation could explain why measures of body excess and fat distribution, which are fundamentally risk factors for metabolic derangements, have exhausted their role.

Sensitivity, specificity, positive likelihood ratio, negative likelihood ratio, positive predictive value and negative predictive value were not different in boys for all indexes, while in girls CMI showed less sensitivity compared to the other two indexes, but the best specificity, best positive predictive values and the best positive likelihood ratio. The positive likelihood ratio of all indexes was found to be useful in boys as a predictor of MetS, while in girls only CMI was useful. The negative likelihood ratio was similar in all subjects.

Consequently, we choose the TG/HDL-C ratio as the simplest index. In our study the TG/HDL-C ratio showed a strong correlation with the clinical and biochemical parameters of MetS, but not with glucose levels. The odds ratio showed a strong association of TG/HDL-C ratio with the high triglycerides and low HDL-C, but not with glucose level, blood pressure and waist circumference. Overall, these data confirmed previous observations, showing that the TG/HDL-C ratio had high sensitivity and specificity for detecting MetS in children and adolescents as well as in adult patients with obesity [39,40,41]. Furthermore, the usefulness of the TG/HDL-C ratio in identifying worsened cardiometabolic profile and preclinical signs of organ damage has been previously reported in both obese and non-obese children [42,43].

Another aim of our study was to verify whether the cutoff of TG/HDL-C ratio for detecting the risk of MetS varied by sex and age. Thus, the trend of the TG/HDL-C ratio was calculated in boys and girls of different ages. In both genders a clear increase over the years was observed (Figure 3), this age-dependent trend requiring to be considered when interpreting the meaning of the calculated index.

Conversely to what was observed in a previous paper by our group, showing that BMI was similar to other indexes in detecting MetS in children and adolescents [15], in the present study BMI proved to be less accurate in identifying MetS than the TG/HDL-C ratio. In our opinion, the reason for this difference could be the choice of the population evaluated, as the SDS BMI was equal to or greater than 3.00 SDS in all patients in the present study, unlike our previous study. The different degree of obesity could explain why BMI, which is actually a risk factor for cardiometabolic diseases and is very reliable in a population of overweight or mild obese subjects, does not show the same accuracy in a population where the metabolic status is already severely compromised and where the anthropometric measures no longer play any prognostic role.

The strength of this study is the large number of subjects enrolled, which results in an adequate statistical analysis. In addition, the recruitment of children and adolescents with severe obesity was made in a single third level center, with the same well-trained medical staff and the same laboratory. However, there are some limitations to our study. Since our investigation is a cross-sectional study, we are unable to predict whether the MetS will persist in our patients or will appear in the future in those patients who are borderline for developing MetS. Another weakness is that the vast majority of patients were Caucasian, and its findings may not be extended to other ethnic groups.

In conclusion, TG/HDL ratio, which just needs the evaluation of two simple biochemical parameters, offers the same accuracy as other more sophisticated indexes in recognizing MetS in children and adolescents with severe obesity, thus resulting as the best predictor that can be easily used.

Finally, our results suggest that changes in the TG/HDL-C ratio over the years should be considered in order to ensure the timely diagnosis of MetS and to set the most age-appropriate therapeutic and rehabilitative approaches [44].

## Figures and Tables

**Figure 1 jcm-11-04488-f001:**
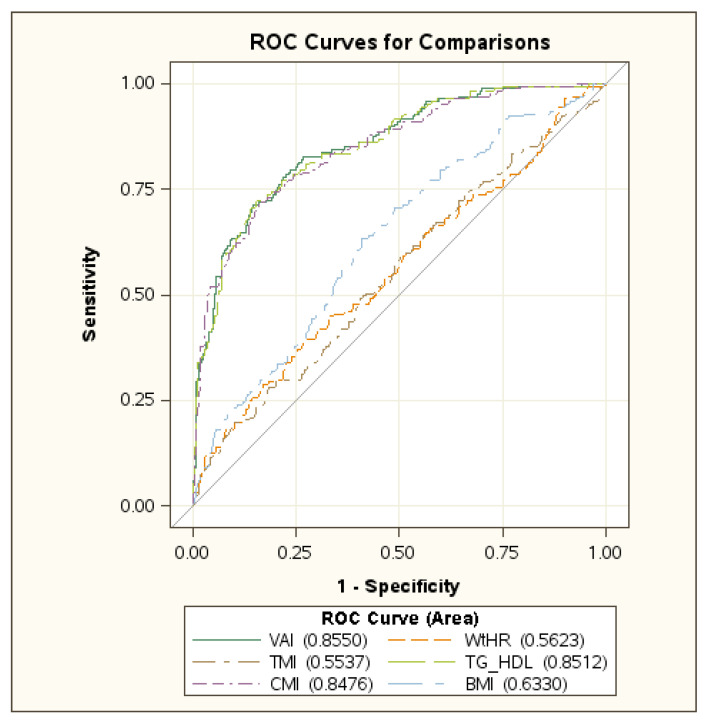
ROC curve in boys. Abbreviations: VAI: Visceral Adiposity Index; WtHR: Waist-to-Height ratio; TMI: Tri-Ponderal Mass Index; TG/HDL: triglyceride to high density-lipoprotein cholesterol ratio; CMI: Cardiometabolic Index; BMI: Body Mass Index.

**Figure 2 jcm-11-04488-f002:**
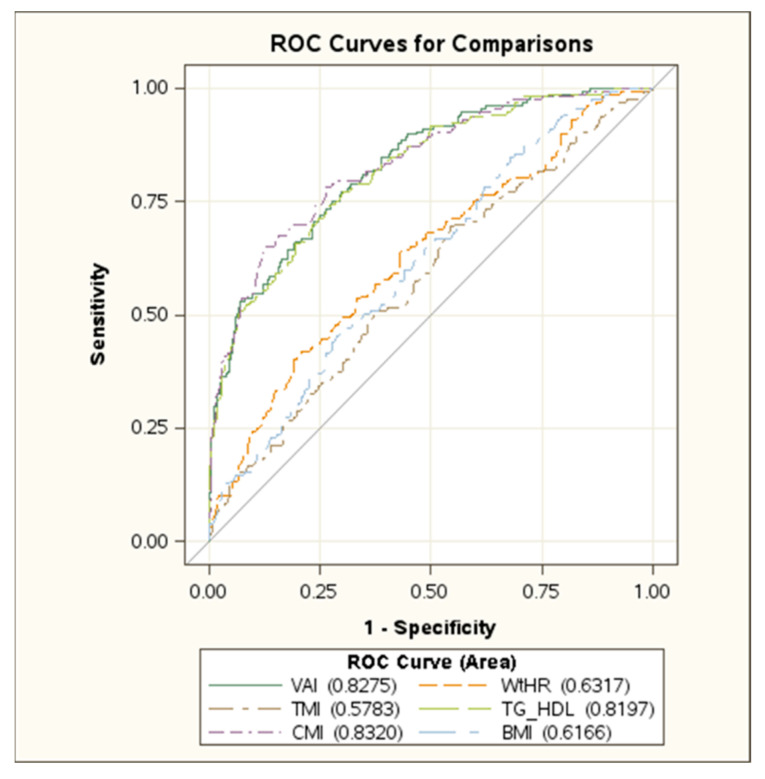
ROC curve in girls. Abbreviations: VAI: Visceral Adiposity Index; WtHR: Waist-to-Height ratio; TMI: Tri-Ponderal Mass Index; TG/HDL: triglyceride to high density-lipoprotein cholesterol ratio; CMI: Cardiometabolic Index; BMI: Body Mass Index.

**Figure 3 jcm-11-04488-f003:**
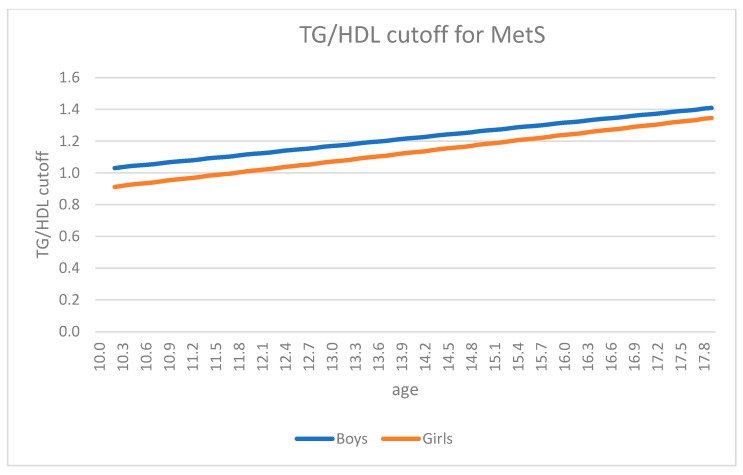
Trend of TG/HDL cutoff for MetS, according to age and sex. The mean difference of the cut-off values calculated at all ages between sexes (*p* < 0.05). Abbreviation: TG/HDL: triglyceride to high density-lipoprotein cholesterol ratio.

**Table 1 jcm-11-04488-t001:** Clinical and biochemical data in boys and girls. Data are expressed as median (interquartile range).

	Total (*n* = 1065)	Boys (*n* = 502)	Girls (*n* = 563)
Age (yr)	14.9 (13–16.4)	14.7 (12.7–16.2) ^&^	15 (13.2–16.5)
Weight (kg)	105.2 (93.1–119.4)	110.5 (93.1–126.9) *	101.8 (93–114)
Height (cm)	1.6 (1.6–1.7)	1.7 (1.6–1.8)	1.6 (1.6–1.7)
BMI (kg/m^2^)	38.8 (36–42.6)	38.3 (35.3–41.8) *	39.3 (36.7–42.8)
BMI SDS	3.5 (3.2–3.7)	3.5 (3.3–3.7)	3.4 (3.2–3.7)
Waist Circumference (cm)	117 (107–126)	120 (110–129) *	115 (106–124)
Glucose (mmol/L)	79 (74–84)	80 (76–84) ^$^	79 (73–83)
Insulin (µU/mL)	14.8 (10–19.9)	15.3 (10–20.1)	14.5 (10–19.7)
HOMA	2.9 (1.9–3.9)	3 (2–4.1)	2.8 (1.9–3.8)
HDL Cholesterol (mg/dL)	42 (36–49)	41 (35–47) *	44 (37–50)
LDL Cholesterol (mg/dL)	104 (86–123)	106.5 (87–126)	102 (84–122)
Triglycerides (mg/dL)	90 (69–119)	93 (71–125) ^&^	88 (68–113)
SBP (mm/Hg)	125 (120–130)	130 (120–140) *	120 (120–130)
DBP (mm/Hg)	80 (70–80)	80 (70–80) ^&^	80 (70–80)
VAI	1.5 (1.1–2.2)	1.4 (0.9–1.9) *	1.7 (1.2–2.4)
WtHR (cm/cm)	0.72 (0.67–0.76)	0.71 (0.67–0.76)	0.72 (0.66–0.77)
TMI (kg/m^3^)	23.8 (22–25.9)	22.7 (21.1–25) *	24.5 (22.8–26.7)
TG/HDL	0.9 (0.7–1.3)	1 (0.7–1.4) *	0.9 (0.6–1.2)
CMI	0.7 (0.5–1)	0.7 (0.5–1) *	0.6 (0.4–0.9)

Abbreviations: BMI: Body Mass Index; BMI SDS: BMI standard deviation score; HDL: high-density lipoprotein; LDL: low-density lipoprotein; SBP: systolic blood pressure; DBP: diastolic blood pressure; VAI: Visceral Adiposity Index; WtHR: Waist-to-Height ratio; TMI: Tri-Ponderal Mass Index; TG/HDL: triglyceride to high density-lipoprotein cholesterol ratio; CMI: Cardiometabolic Index. For the difference between boys and girls: * *p* < 0.0001; ^$^ *p* < 0.005; ^&^ *p* < 0.05.

**Table 2 jcm-11-04488-t002:** Clinical and biochemical data subdivided according to the presence or absence of MetS. Data are expressed as median (interquartile range).

	MetS	No MetS
Number of subjects	741	324
Age (yr)	15.8 (14.2–16.8) *	14.5 (12.6–16)
Weight (kg)	114.4 (100.8–130.6) *	101.2 (90.1–115.8)
Height (cm)	1.7 (1.6–1.7) *	1.6 (1.6–1.7)
BMI (kg/m^2^)	40 (37.3–43.9) *	38.3 (35.7–41.7)
BMI SDS	3.5 (3.3–3.8)	3.4 (3.2–3.7)
WC (cm)	123 (113–133) *	115 (106–124)
Glucose (mmol/L)	79 (75–84)	79 (74–84)
Insulin (µU/mL)	17.1 (13.1–22.8) *	13.4 (9.3–18.7)
HOMA	3.3 (2.5–4.6) *	2.6 (1.8–3.7)
HDL Cholesterol (mg/dL)	35 (32–38.5) *	46 (41–51)
LDL Cholesterol (mg/dL)	108 (86.5–127) ^&^	102 (86–123)
Triglycerides (mg/dL)	117 (88–157) *	83 (66–105)
SBP (mm/Hg)	130 (130–140) *	120 (120–130)
DBP (mm/Hg)	80 (80–90) *	80 (70–80)
VAI	2.3 (1.7–3.3) *	1.3 (0.9–1.7)
WtHR (cm/cm)	0.73 (0.68–0.79) *	0.71 (0.67–0.75)
TMI (kg/m^3^)	24.1 (22.2–26.7) ^$^	23.6 (21.9–25.8)
TG/HDL	1.5 (1.1–2.1) *	0.8 (0.6–1.1)
CMI	1.1 (0.8–1.5) *	0.6 (0.4–0.8)

Abbreviations: MetS: Metabolic Syndrome; BMI: Body Mass Index; BMI SDS: BMI standard deviation score; WC: Waist Circumference; HDL: high-density lipoprotein; LDL: low-density lipoprotein; SBP: systolic blood pressure; DBP: diastolic blood pressure; VAI: Visceral Adiposity Index; WtHR: Waist-to-Height ratio; TMI: Tri-Ponderal Mass Index; TG/HDL: triglyceride to high density-lipoprotein cholesterol ratio; CMI: Cardiometabolic Index. For the difference between MetS and No Mets: * *p* < 0.0001; ^$^ *p* < 0.01; ^&^ *p* < 0.05.

**Table 3 jcm-11-04488-t003:** Sensitivity, specificity, positive likelihood ratio (PLR), negative likelihood ratio (NLR), positive predictive value (PPV), and negative predictive value (NPV) of VAI (**a**), CMI (**b**) and TG/HDL-Cholesterol (**c**).

(a)		
	Boys	Girls
Sensitivity	0.713 (0.673–0.752)	0.752 (0.716–0.787)
Specificity	0.854 (0.823–0.885)	0.724 (0.687–0.761)
PPV	0.708 (0.669–0.748)	0.513 (0.472–0.554)
NPV	0.856 (0.826–0.887)	0.883 (0.856–0.909)
PLR	4.9	2.7
NLR	0.3	0.3
**(b)**		
	**Boys**	**Girls**
Sensitivity	0.719 (0.679–0.758)	0.65 (0.61–0.689) *
Specifity	0.836 (0.803–0.868)	0.872 (0.844–0.9) *
PPV	0.686 (0.645–0.726)	0.662 (0.623–0.701) *
NPV	0.856 (0.826–0.887)	0.866 (0.837–0.894)
PLR	4.4	5.1 *
NLR	0.3	0.4
**(c)**		
	**Boys**	**Girls**
Sensitivity	0.725 (0.685–0.764)	0.745 (0.709–0.781)
Specifity	0.845 (0.813–0.876)	0.724 (0.687–0.761)
PPV	0.699 (0.659–0.74)	0.511 (0.47–0.552)
NPV	0.86 (0.83–0.891)	0.88 (0.853–0.907)
PLR	4.7	2.7
NLR	0.3	0.4

Abbreviations: VAI: Visceral Adiposity Index; CMI: Cardiometabolic Index; TG/HDL: triglyceride to high density-lipoprotein cholesterol ratio. For significance: * *p* < 0.05 compared to the other indices.

**Table 4 jcm-11-04488-t004:** Bivariate correlation coefficients between TG/HDL and metabolic characteristics.

	Total	Boys	Girls
Weight (kg)	0.25 *	0.25 *	0.08
Height (cm)	0.19 *	0.19 *	0.04
BMI (kg/m^2^)	0.22 *	0.22 *	0.08
WC (cm)	0.23 *	0.23 *	0.17 *
Glucose (mmol/L)	−0.01	0.00	0.04
HDL Cholesterol (mg/dL)	−0.68 *	−0.67 *	−0.70 *
LDL Cholesterol (mg/dL)	0.22 *	0.22 *	0.42 *
Triglycerides (mg/dL)	0.89 *	0.89 *	0.90 *
SBP (mm/Hg)	0.11 ^&^	0.11 ^&^	0.10 ^&^
DBP (mm/Hg)	0.08	0.08	0.05
BMI SDS	0.16 ^$^	0.16 ^$^	0.08

Abbreviations: BMI: Body Mass Index; WC: Waist Circumference; HDL: high-density lipoprotein; LDL: low-density lipoprotein; SBP: systolic blood pressure; DBP: diastolic blood pressure; SDS: standard deviation score. For significance: * *p* < 0.0001; ^$^ *p* < 0.001; ^&^ *p* < 0.05.

**Table 5 jcm-11-04488-t005:** Odds ratio and 95% confidence interval for cardiometabolic risk factors across TG/HDL adjusted for age.

	Boys		Girls	
	OR (95% CI)	Pseudo R Square	OR (95% CI)	Pseudo R Square
Large WC (IDF)	n.c. *	n.c. *	267.1 (0.1 to infinite)	0.138
High Glucose (IDF)	1.7 (0.5–6.1)	0.025	1.8 (0.6–5.2)	0.034
Low HDL cholesterol (IDF)	17.1 (9.8–29.9) ^&^	0.43	22.9 (12.4–42.2) ^&^	0.456
High Triglycerides (IDF)	121.2 (39.6–370.6) ^&^	0.681	167.3 (50.3–556.9) ^&^	0.759
High Blood Pressure (IDF)	0.9 (0.6–1.2)	0.19	1.3 (1–1.7)	0.056

* Not calculated. All observations have the same response. Abbreviation: WC: Waist Circumference; IDF: International Diabetes Federation criteria; HDL: high-density lipoprotein; TG/HDL: triglyceride to high density-lipoprotein cholesterol ratio. For significance: ^&^ *p* < 0.0001.

## Data Availability

Raw data are available upon a reasonable request to the corresponding author or will be available on Zenodo.org after the Authors will have the definitive doi number of the paper available.

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
