# Peer review of "High Tg/HDL-Cholesterol Ratio Highlights a Higher Risk of Metabolic Syndrome in Children and Adolescents with Severe Obesity"

_jcm, 2022, doi:10.3390/jcm11154488_

Round 1

Reviewer 1 Report

This is an interesting study comparing several indices to detect
metabolic syndrome. One strength of the manuscript is the large   sample size and comparison of many indices. Overall, the manuscript   is well written and the results are interesting and should be published.

I have some problems to understand the analysis and ask the authors if this can be clarified.

1) line 158: A quantile regression is used to model body composition index as a function of age. I cannot find the results of this analysis.

2) I assume that the significance test in table 1 compare boys and
      girls? Please clarify this in the caption.

3) In table 2 the significance tests compare MetS and no MetS?

4) The ROC analysis does not reveal information about the best
   cut-off values and the corresponding sensitivity and specificity. Can you provide this information?

5) I don't understand the analysis in supp table S1: What is shown
     in the three tables? What contrast method is used? Looks like a
      Dunnet contrast was applied? Alpha error adjustment?

6) How was data in table 5 analysed? Logistic Regression? What pseudo R^2 was used?

7) Which method was used in figure 3? A linear regression with    response TG/HDL-C ratio? How were the age related cut-off values
computed? Do you tested for interaction age X gender?

Author Response

Comments and Suggestions for Authors

This is an interesting study comparing several indices to detect
metabolic syndrome. One strength of the manuscript is the large   sample size and comparison of many indices. Overall, the manuscript   is well written and the results are interesting and should be published.

I have some problems to understand the analysis and ask the authors if this can be clarified.
Thank you for taking the time to review our article carefully. Your feedback was very helpful in improving the overall quality of the manuscript. Here enclosed please find a point by point response to your queries below.

Q1. line 158: A quantile regression is used to model body composition index as a function of age. I cannot find the results of this analysis.

A1. The percentile of the ordered values corresponding to the optimal cutoff value identified with the Youden index, was used as regression quantile. The age-predictive values resulting from the regression are represented in the cutoff graph for the Mets. The statistics for regressions are reported in the following boxes (for boys and girls).

Boys TG/HDL

Summary Statistics

Variable

Q1

Median

Q3

Mean

Standard
Deviation

MAD

Age

12.6600

14.7050

16.1600

14.4527

2.1107

2.5278

TG_HDL

0.7173

1.0310

1.4346

1.1597

0.6380

0.5226

Quantile and Objective Function

Quantile

0.6573705

Objective Function

115.2791

Predicted Value at Mean

1.2404

Parameter Estimates

Parameter

DF

Estimate

Intercept

1

0.5334

Age

1

0.0489

Test Results

Test

Test Statistic

DF

Chi-Square

Pr > ChiSq

Wald

8.2419

1

8.24

0.0041

Likelihood Ratio

7.9382

1

7.94

0.0048

Girls TG/HDL

Summary Statistics

Variable

Q1

Median

Q3

Mean

Standard
Deviation

MAD

Age

12.6600

14.7050

16.1600

14.4527

2.1107

2.5278

TG_HDL

0.7173

1.0310

1.4346

1.1597

0.6380

0.5226

Quantile and Objective Function

Quantile

0.5932504

Objective Function

116.5757

Predicted Value at Mean

1.1526

Parameter Estimates

Parameter

DF

Estimate

Intercept

1

0.3424

Age

1

0.0561

Test Results

Test

Test Statistic

DF

Chi-Square

Pr > ChiSq

Wald

13.4575

1

13.46

0.0002

Likelihood Ratio

10.3072

1

10.31

0.0013

Boys VAI

Summary Statistics

Variable

Q1

Median

Q3

Mean

Standard
Deviation

MAD

Age

12.6600

14.7050

16.1600

14.4527

2.1107

2.5278

VAI

0.9408

1.3793

1.9300

1.5751

0.8884

0.7139

Quantile and Objective Function

Quantile

0.669323

Objective Function

158.6746

Predicted Value at Mean

1.7137

Parameter Estimates

Parameter

DF

Estimate

Intercept

1

0.5325

Age

1

0.0817

Test Results

Test

Test Statistic

DF

Chi-Square

Pr > ChiSq

Wald

10.8939

1

10.89

0.0010

Likelihood Ratio

10.7044

1

10.70

0.0011

Girls VAI

Summary Statistics

Variable

Q1

Median

Q3

Mean

Standard
Deviation

MAD

Age

12.6600

14.7050

16.1600

14.4527

2.1107

2.5278

VAI

0.9408

1.3793

1.9300

1.5751

0.8884

0.7139

Quantile and Objective Function

Quantile

0.589698

Objective Function

160.4864

Predicted Value at Mean

1.5268

Parameter Estimates

Parameter

DF

Estimate

Intercept

1

0.3086

Age

1

0.0843

Test Results

Test

Test Statistic

DF

Chi-Square

Pr > ChiSq

Wald

15.3390

1

15.34

<.0001

Likelihood Ratio

14.6530

1

14.65

0.0001

BOYS CMI

Summary Statistics

Variable

Q1

Median

Q3

Mean

Standard
Deviation

MAD

Age

12.6600

14.7050

16.1600

14.4527

2.1107

2.5278

CMI

0.5005

0.7493

1.0436

0.8416

0.4802

0.3938

Quantile and Objective Function

Quantile

0.6533865

Objective Function

87.4906

Predicted Value at Mean

0.8879

Parameter Estimates

Parameter

DF

Estimate

Intercept

1

0.3406

Age

1

0.0379

Test Results

Test

Test Statistic

DF

Chi-Square

Pr > ChiSq

Wald

7.6504

1

7.65

0.0057

Likelihood Ratio

11.0503

1

11.05

0.0009

GIRLS CMI

Summary Statistics

Variable

Q1

Median

Q3

Mean

Standard
Deviation

MAD

Age

12.6600

14.7050

16.1600

14.4527

2.1107

2.5278

CMI

0.5005

0.7493

1.0436

0.8416

0.4802

0.3938

Quantile and Objective Function

Quantile

0.7282416

Objective Function

83.5634

Predicted Value at Mean

1.0082

Parameter Estimates

Parameter

DF

Estimate

Intercept

1

0.2922

Age

1

0.0495

Test Results

Test

Test Statistic

DF

Chi-Square

Pr > ChiSq

Wald

10.8086

1

10.81

0.0010

Likelihood Ratio

8.0041

1

8.00

0.0047

Q2. I assume that the significance test in table 1 compare boys and girls? Please clarify this in the caption.

A2. Your interpretation was correct. We have added a sentence to clarify this point.

Q3. In table 2 the significance tests compare MetS and no MetS?

A3. Your interpretation was correct. We have added a sentence to clarify this point.

Q4. The ROC analysis does not reveal information about the best cut-off values and the corresponding sensitivity and specificity. Can you provide this information?

A4. Cut offs identified with ROC analysis were the following:

Boys

Girls

VAI

1.71

1,89

TG/HDL

1.26

0.99

CMI

0.89

0.87

Sensitivity and other indexes are reported in table 3.

Q5. I don't understand the analysis in supp table S1: What is shown in the three tables? What contrast method is used? Looks like a Dunnet contrast was applied? Alpha error adjustment?

A5. The analysis compared the different ROC models (in terms of AUC) and produced a contrast matrix of differences between each ROC curve and the reference curve (VAI, TG/HDL, CMI). Differences were tested with chi-square test and the 95% Wald confidence limits of the differences was showed.

Q6. How was data in table 5 analyzed? Logistic Regression? What pseudo R^2 was used?

A6. Data were analyzed by logistic regression. We apologize for the typing error: R square was actually used. In the revised table the corrected values have been reported.

Boys

Girls

OR (95% CI)

R square

OR (95% CI)

R square

Large WC (IDF)

n.c.*

n.c.*

267.1 (0.1 to infinite)

0.006

High Glucose (IDF)

1.7 (0.5-6.1)

0.002

1.8 (0.6-5.2)

0.003

Low HDL cholesterol (IDF)

17.1 (9.8-29.9)&

0.321

22.9 (12.4-42.2)&

0.341

High Triglycerides (IDF)

121.2 (39.6-370.6)&

0.351

167.3 (50.3-556.9)&

0.374

High Blood Pressure (IDF)

0.9 (0.6-1.2)

0.141

1.3 (1-1.7)

0.042

Table 5. Odds ratio and 95% confidence interval for cardiometabolic risk factors across TG/HDL adjusted for age.

Q7. Which method was used in figure 3? A linear regression with response TG/HDL-C ratio? How were the age related cut-off values computed? Do you tested for interaction age X gender?

A7. The percentile of the ordered values corresponding to the optimal cutoff value identified with the Youden index, was used as regression quantile. The age-predictive values resulting from the regression have been represented in the cutoff graph for the Mets. Age rounded to first decimal place and separated analysis by gender.

Reviewer 2 Report

 The manuscript “High Tg/HDL-cholesterol ratio highlights a higher risk of metabolic syndrome in children and adolescents with severe obesity” is well written. The authors used extensive clinical and paraclincal data that are well documented. I think that the research question is relevant to the field, yet it can be better addressed in the methodology section. The conclusions are consistent with the evidence and arguments presented and I consider the references appropriate.

Suggestions to the authors:

The manuscript needs a few language corrections before being published (revision by a native English speaker is strongly advised).

 “..offers the same accuracy of other more sophisticated”- I suggest “…as other”

 “For this reason, the need to have available tools allowing early identification of patients with high degree of obesity at risk for MetS has become mandatory.” I think the phrase needs to be rephrased

I would suggest “unnecessary investigations”

WHR- please use the abbreviations consistently

ABSI- define the terms at the first use in text

“The data were first scrutinized for outliers, using a cutoff of 4.5 SDS”.- please be more specific to which variable this cut-off refers to

“In Table 4 the correlation between TG/HDL-C ratio with the”…. I suggest “The correlation between… and…”

“In Table 5 the odds ratio between TG/HDL-C ratio with the different cardiometabolic  risk factors are reported. In both sexes, a strong association with high triglycerides and low HDL-C was found, but not with glucose and blood pressure”.- the phase should de reformulated and the data better explained in text; the findings seem redundant

“Figure 5 Trend of TG/HDL cutoff for MetS, according to age and sex”- it is not very clear to which cut off are you referring to; what was the value of this cutoff? is there a significant difference between genders?

I suggest a more clear and comprehensive expression of your results in text

As both TG and HDL-C are included in the definition of MetS, the association between the two, or their ratio, and the presence of MetS is more than obvious from the beginning, and the demonstration may seem redundant; maybe the association between TG/HDL and cardiovascular risk should have been of more value

“Conversely to what observed in a previous paper by our group, showing that BMI  was similar to other indexes in detecting MetS in children and adolescents (15), in the  present study BMI resulted less accurate in identifying MetS than the TG/HDL-C ratio.”- as I already mentioned before both TG and HDL are already included in the definition of MetS

Author Response

The manuscript “High Tg/HDL-cholesterol ratio highlights a higher risk of metabolic syndrome in children and adolescents with severe obesity” is well written. The authors used extensive clinical and paraclincal data that are well documented. I think that the research question is relevant to the field, yet it can be better addressed in the methodology section. The conclusions are consistent with the evidence and arguments presented and I consider the references appropriate.

Thank you for taking the time to review our article carefully. Your feedback was very helpful in improving the overall quality of the manuscript. Here enclosed please find a point by point response to your queries below.

Q1. The manuscript needs a few language corrections before being published (revision by a native English speaker is strongly advised).

 “..offers the same accuracy of other more sophisticated”- I suggest “…as other”

 “For this reason, the need to have available tools allowing early identification of patients with high degree of obesity at risk for MetS has become mandatory.” I think the phrase needs to be rephrased

I would suggest “unnecessary investigations”

A1. The paper has been revised by an English teacher.

Q2. WHR- please use the abbreviations consistently

ABSI- define the terms at the first use in text

A2. The requested corrections have been done.

Q3. “The data were first scrutinized for outliers, using a cutoff of 4.5 SDS”.- please be more specific to which variable this cut-off refers to

A3. The cut-offs were referred to VAI, TG/HDL, CMI. We have clarified this point in the text

Q4. “In Table 4 the correlation between TG/HDL-C ratio with the”…. I suggest “The correlation between… and…”

A4. Corrected as suggested

Q5. “In Table 5 the odds ratio between TG/HDL-C ratio with the different cardiometabolic  risk factors are reported. In both sexes, a strong association with high triglycerides and low HDL-C was found, but not with glucose and blood pressure”.- the phrase should be reformulated and the data better explained in text; the findings seem redundant

 A5. The sentence has been reformulated

Q6. “Figure 5 Trend of TG/HDL cutoff for MetS, according to age and sex”- it is not very clear to which cut off are you referring to; what was the value of this cutoff? is there a significant difference between genders?

A6. The trend in Figure 3 (not fig 5!) shows the cut-off values at various ages calculated with a quantile regression starting from the percentile of the cut-off identified with the ROC curve. The mean difference of the cut-off values between sexes, calculated at all ages, was always significant (p<0.05). We have added a sentence to clarify this point

Q7. I suggest a more clear and comprehensive expression of your results in text.

As both TG and HDL-C are included in the definition of MetS, the association between the two, or their ratio, and the presence of MetS is more than obvious from the beginning, and the demonstration may seem redundant; maybe the association between TG/HDL and cardiovascular risk should have been of more value

A7. The reviewer is correct, in fact in table 4 and 5 the correlations and the OR between TG/HDL and the cardiovascular risk factors are reported.

Q8. “Conversely to what observed in a previous paper by our group, showing that BMI  was similar to other indexes in detecting MetS in children and adolescents (15), in the  present study BMI resulted less accurate in identifying MetS than the TG/HDL-C ratio.”- as I already mentioned before both TG and HDL are already included in the definition of MetS

A8. It is correct that TG and HDL-C are included in the definition of MetS, as for other indices. However, the aim of this study was to find out the simplest and more accurate index, among all indices, able to detect MetS in a particular population of extremely obese patients